# Characteristics of the Mating Behavior of Domesticated Geese from *Anser cygnoides* and *Anser anser*

**DOI:** 10.3390/ani12182326

**Published:** 2022-09-07

**Authors:** Qiang Bao, Yang Zhang, Ying Yao, Xuan Luo, Wenming Zhao, Jiwen Wang, Guohong Chen, Qi Xu

**Affiliations:** 1Key Laboratory for Evaluation and Utilization of Poultry Genetic Resources of Ministry of Agriculture and Rural Affairs, Yangzhou University, Yangzhou 225009, China; 2Farm Animal Genetic Resources Exploration and Innovation Key Laboratory of Sichuan Province, Sichuan Agricultural University, Chengdu 625014, China; 3Joint International Research Laboratory of Agriculture and Agri-Product Safety, The Ministry of Education of China, Yangzhou University, Yangzhou 225009, China

**Keywords:** goose, mating behavior, fertilization rate, reproduction

## Abstract

**Simple Summary:**

Mating behavior is a critically important component for producing the offspring of poultry. Geese are important poultry with high economic value, but studies showing the characteristic of mating behavior are limited. On this basis, the research investigated the relationship between mating behavior and reproductive performance of 300-day-old Sichuan white geese (*Anser cygnoides*), Zhedong white geese *(Anser cygnoides*), and Hungarian geese (*Anser anser*). The results showed that different mating behaviors among different breeds were mainly reflected in time preference and successful copulation frequency. These were not only helpful in improving the understanding mating process of domesticated geese from *Anser cygnoides* and *Anser anser*, but also might contribute to improving the frequency of successful copulation according to characteristics of the mating behavior.

**Abstract:**

Mating behavior is a critically important component of poultry reproduction. Here, a total of 135 geese were selected, specifically, Sichuan white geese (*Anser cygnoides*), Zhedong white geese (*Anser cygnoides*), and Hungarian geese (*Anser anser*) (300-day-old), and the mating behavior was monitored daily from 6:00 a.m. to 6:00 p.m. during the 20-day observation period. The results showed that the mating process included mounting, female cooperation, and successful copulation. Overall, the three breeds preferred mating on land. More than thirty percent of the mating time was primarily concentrated from 4:00 p.m. to 6:00 p.m. in domesticated geese from *Anser cygnoides*, the corresponding values for Sichuan white geese and Zhedong white geese were 32.0% and 33.3%, respectively. The mating of the Hungarian geese usually took place in the morning. In addition, the frequency of successful copulation of Sichuan white geese and Zhedong white geese were 2.31 and 1.94 times per day, significantly greater than that of Hungarian geese (0.89 times). Furthermore, a significant positive correlation between successful copulation and laying rates (*r* = 0.985) or fertilization rates (*r* = 0.992) was observed in Hungarian geese. Taken together, the mating behaviors among the different breeds were mainly reflected in time preference and successful copulation frequency.

## 1. Introduction

The mating behavior of animals is necessary for producing offspring, and each species has its mating system in course of its evolution [1,2]. Similar to mammals, the mating behavior of poultry is not only an essential guarantee for reproduction, but also crucial to attaining successful fertilization. Typically, the reproductive success of poultry also depends on sexual interaction and mating behavior of potential mates in a competitive social structure [3,4]. The previous study by Zhang et al. suggested that sexual behavior was the main factor affecting the reproductive performance of poultry [5]. In addition, mating behavior seems to correlate with hormonal regulation and the time after oviposition [6,7,8]. Therefore, females’ reproductive state is also closely related to mating behavior.

China is the world’s largest goose farming country and has a wide variety of goose resources. In recent years, Hungarian geese have been widely raised as an excellent introduced breed, and their popularity was increasing in many regions. However, geese are accompanied by various different reproductive performances among breeds. In general, the rate of fertilization and hatchability, as the essential component of reproduction performance are important economic traits in geese production and are affected by several factors, such as sex ratio, sexual behavior, nutrition, housing system, and the health of the birds [9,10]. Previous studies have reported that male geese complete most mating rituals in the first half of the breeding season, including successful copulation, whereas female geese are more likely to achieve a desirable fertilization rate during the entire reproduction season [6,11]. Successful copulation is a critical element for the efficient production of geese in the natural mating system [12]. Fertilization rate and hatchability in commercial flocks may be associated with the intensity of sexual interaction, and the frequency of successful copulation positively impacts the probability of egg fertilization [5]. On average, Chinese geese successfully mate 2.5–5.5 times daily [5], compared with 1.2–5.5 times daily for Zatorska geese [13]. However, those studies were mainly focused on copulation frequency for mating behavior differences in various breeds and seldom concentrated on other aspects of mating behavior.

Although a few studies have found that geese exhibited the characteristic mating behavior [14], the relationship between mating behavior and reproductive performance of domesticated geese from *Anser cygnoides* and *Anser anser* is currently unclear. In this study, two Chinese goose breeds (Sichuan white geese and Zhedong white geese, *Anser cygnoides*) and one European goose breed (Hungarian geese, *Anser anser*) were selected, and their preferred mating time, location, and frequency were investigated. It may provide valuable information to characterize mating behavior and confirm the importance of mating behavior for reproduction in geese.

## 2. Materials and Methods

### 2.1. Ethics Approval

All bird-handling protocols were approved by the Ethics Committee on Animal Experiments of Yangzhou University (Permit Number: YZUDWSY, Government of Jiangsu Province, China). Any experimentation involving geese also adhered to the Regulations for the Administration of Affairs Concerning Experimental Animals, approved by the State Council of the People’s Republic of China.

### 2.2. Experimental Design and Facilities

The experiment was carried out at the National Waterfowl Germplasm Resources Pool (Taizhou, China). The three goose breeds, including Sichuan white geese (Anser cygnoides), Zhedong white geese (Anser cygnoides), and Hungarian geese (Anser anser), were raised separately under the same feeding and management regime before the start of the experiment. To avoid meeting other members of opposite sexes before the study, geese were separated according to sex and raised separately. According to egg production records, the three goose breeds start their first egg at an average age of 210-day-old, and the study was carried out when the geese were already reproductively mature. During the observation period, the geese were raised in 9 separate pens (3 biological replicates/breed, (3♂ + 12♀/pen). Geese stayed together with opposite sexes when they were released to an outdoor area of the pen at 300-days-old, which included a 2.5 × 5.0 m playground and 2.5 × 4.0 m water surface. Moreover, the indoor areas of each pen had a roof, and were equipped with one laying nest (96 cm × 85 cm). The bottom of the nests was covered with rice husks. The nests were large enough to comfortably hold several standing geese and also allowed for the geese to turn around easily. All pens were in one shed with cement-type flooring, two trough-type feeders were equipped in the indoor area, and one bell-type drinker was placed in the playground of the outdoor area. We used a total of 27 cameras to record the behavior of the different goose breeds. In order to eliminate the influence of the blind area and ensure accurate recording of behaviors, three infrared video cameras (Jindun, 720 p, resolution: 1280 × 960, frame Rate: 25 frame/s, Nanjing, China) were set up above water surfaces and land areas in the pen of each group. Hikvision digital recorder (720 p, 32 channels, resolution: 1280 × 960, 2.4 TB, Shenzhen, China) was used for recording the videos.

All geese were provided with the same diet during the experimental period (crude protein: 16%, metabolizable energy: 11.29 MJ/kg), combined with coarse and concentrated material (Table 1). All geese were given food and water ad libitum. The experiment was performed at 22 °C;/13 °C; (day/night), on a 15L: 9D natural lighting photoperiod.

### 2.3. Date Collection

The mating behaviors of all geese were recorded for 12 h per day from 6:00 a.m. to 6:00 p.m. over the 20-day observation period. In addition, mating behavior was additionally evaluated in 5-day phases: days 6–10, 11–15, and 16–20 after females and males were mixed. The time, location, and frequency of behaviors were recorded by continuous sampling from video. Due to the large volume of data, the video records are stored as backup and collected on a computer daily during the observation period. Then, the mating behaviors in all videos were observed and evaluated according to video records by an individual researcher to minimize variation. The same observation index and evaluation criteria were applied the same way to all the videos. For identification of female geese, the feathers of all 108 females were marked with colored paint. Human activities inside and outside the pens were minimized during the observation period to avoid disturbing the geese. Furthermore, the mating process consists of at least three behaviors: mounting, female cooperation, and copulation success. The description of the criteria for the mating behavior can be seen in Table 2. The average frequency per day was calculated by counting the number of occurrences of mating behaviors during the observation period.

Throughout the 20-day observation period, eggs were collected, and laying rates were calculated for each pen. After disinfection, settable eggs were temporarily stored at 13 °C and 75% relative humidity. The Sanyuan incubator manufacturer (SY3872, Bengbu, China) was used for hatching. Before hatching, the eggs were cleaned and sterilized with 70% ethanol. Then the eggs were incubated under standard conditions. All eggs were turned every two hours until 28 days. The incubation temperature was controlled at 38, 37.5, and 36.5 °C; during days 1 to 14, 15 to 28, and 29 to 31. Meanwhile, the humidity was controlled at 65–70%, 60–65%, 65–70%, and 72% during days 1 to 9, 10 to 18, 19 to 28, and 29 to 31, respectively. Moreover, the incubator should be opened twice a day for 35 min each time after the seventeenth day of incubation until all eggs have hatched, and the eggshells’ temperature was regulated at 30–35 °C. After 9 days of incubation, eggs were examined by candling. Eggs containing embryos developed normally, and the blood vessels were radially distributed and appeared bright red. In contrast, eggs containing dead embryos appeared light in color and had irregular blood arcs. All eggs containing embryos were recorded, and the average fertilization rate was calculated based on the statistics.

### 2.4. Statistical Analysis

Characteristics of the mating time and location in different goose breeds were presented using descriptive statistics such as percentages. Statistical analysis was performed using SPSS statistical software (*SPSS 22*, IBM *SPSS* software, Armonk, NY, USA). All data were checked for normality and homogeneity of variance before being analyzed. Relationships between breeds and copulatory behavior were investigated using general linear mixed models (*GLMM*) for continuous normally distributed data, and the data were submitted to the analysis of variance using generalized linear mixed models (*GLIMM*) when the assumptions of normality and homogeneity of variance of the variables were not met. The statistical model included breed (Sichuan white geese vs. Zhedong white geese vs. Hungarian geese) as the main effect and pen (1 to 9) as the random effect. The average laying rate and fertilization rate were analyzed by the Chi-square test. The Pearson correlation was used to assess the relationship between fertilization rate, laying rate, and successful copulation. Results are shown as means with corresponding standard errors (*SEM*). The criterion for statistical significance was set at *p* < 0.05.

## 3. Results

### 3.1. Mating Time and Location in the Different Goose Breeds

In this study, the distributions of mating time and location were observed, in which the preferred mating times of Sichuan white geese and Zhedong white geese were different from those noted in Hungarian geese (Figure 1A). The mating times of Sichuan white geese (32.0%) and Zhedong white geese (33.3%) were primarily concentrated from 4:00 to 6:00 p.m., whereas Hungarian geese (35.2%) mating was performed in the morning between 7:00 to 9:00 a.m. In addition, mating in the water in Sichuan white geese and Zhedong white geese accounted for only 4.81% and 2.05%, respectively, while a relatively high percentage (24%) occurred in the water for Hungarian geese (Figure 1B).

### 3.2. Comparison of Mating Frequencies in Different Goose Breeds

The typical mating behavior of different goose breeds was also monitored in real-time daily (Figure 2). After statistical calculations, the frequencies of mounting (*p* = 0.043) and female cooperation (*p* = 0.031) occurred much more frequently in Sichuan white geese and Zhedong white geese than those in Hungarian geese (Table 3). Moreover, the frequency of successful copulation of Hungarian geese was approximately 0.89 times/day, significantly less than Sichuan white geese (2.31 times/day) and Zhedong white geese (1.94 times/day, *p* = 0.030).

### 3.3. Effects of Successful Copulation on Laying and Fertilization Rate

The laying and fertilization rates were also investigated over the 20 days laying period (Table 4). Among these three goose breeds, the laying rates were greater in Sichuan white geese in comparison with the two other breeds (*p* = 0.002). Moreover, the fertilization rate of Hungarian geese was approximately 66.84%, significantly lower than Sichuan white geese (96.92%) and Zhedong white geese (84.63%) (*p* = 0.005). Next, we examined the correlations between mating behavior and laying or fertilization rate. In this study, there were significant positive correlations between successful copulation and laying or fertilization rates in Hungarian geese (*r* = 0.985, *p* = 0.032; *r* = 0.992, *p* = 0.021, respectively), and no significant correlation was observed in Sichuan white geese and Zhedong white geese.

### 3.4. Effects of the Time Spent Together with Opposite Sexes on Successful Copulation

The successful copulation of Zhedong white geese and Sichuan white geese was observed on days 6–10, 11–15, and 16–20 after they were mixed (Figure 3). The results showed that the successful copulation of Zhedong white geese was significantly greater after the geese stayed together for 10 days (*p* = 0.041), whereas this had a negligible effect on the successful copulation of Sichuan white geese (*p* = 0.359). Due to the low mating frequency of Hungarian geese, they were exempt from this analysis.

## 4. Discussion

In this study, we found that the preferred mating times in the different goose breeds were not consistent during the observation period. A previous study has shown that the mating time of domestic goose was primarily concentrated from 6:00 a.m. to 6:00 p.m. [12]. Our observations revealed that Sichuan white geese and Zhedong white geese primarily concentrated in the afternoon, whereas Hungarian geese mated in the morning. This was similar to the result of Gillette, who found that two-thirds of the mating behavior occurred before noon for Hungarian geese [12]. In addition, these differences might be associated with the female hormone responsiveness to the preferred partner, and the male’s facultative potential to respond to her readiness to breed [15,16]. A previous study also found that geese showed the lesser number of mating when the breeders entered the pen [12]. It suggested that breeders should minimize human activities based on the mating peaks of different goose breeds, including feeding the diet and cleaning the house. Moreover, the lack of reliability testing might limit the accuracy of our outcome observation [17,18]. Therefore, the intra-rater reliability score will be conducted as a future study direction.

For geese, as one of the most economically important waterfowl around the world, land and water surface are important parts of the traditional rearing environment [19]. Currently, poultry husbandry is characterized by intensive husbandry systems with high numbers of geese kept at high stocking densities [20]. Moreover, for water pollution control, water surfaces have been greatly reduced for the intensive rearing of geese. In terms of mating location, all these three goose breeds expressed a preference for mating on land. Another experiment also noted that high fertilization rates are sustained under waterless conditions, confirming their capacity to mate properly on land [4]. However, considering the waterless condition might increase the risk of phallus pecking and phallus hygiene problems, an enhanced approach to the management of geese should be required.

Furthermore, mating behavior was determined in terms of mating frequency and successful copulation, and the variation in successful copulation is the greatest determinant of the reproduction [21]. In this study, the frequencies of mating behaviors were distinctly different in various goose breeds. Sichuan white geese and Zhedong white geese showed a higher frequency of successful copulation. Over 95% of mounting behavior converted to successful copulation in Sichuan white geese and Zhedong white geese. In contrast, we recorded a lower rate of successful copulation in Hungarian geese. A previous study found that the majority of copulations were initiated by male geese [12]. In our study, the frequencies of mounting occurred less frequently in Hungarian geese. Therefore, this phenomenon might be due to fewer male geese engaging in mounting, and failure in effectively converting mounts into successful copulation. On the other hand, female Hungarian geese may also be uncooperative, which has been previously established as a reason for low copulation success [22]. Moreover, several studies have found that the degree of female cooperation could be reflective of her willingness to accept a specific mate [4,23]. Consequently, eliminating aggressive male and uncooperative female geese before forming the groups may significantly improve the frequency of successful copulation. In addition, the reason for the differences between our results and those of previous studies could be attributable to the different breeds used in experiments [4]. Gumulka and Rozenboim also found that the sexual interactions in the mean range (0.4–0.8) from 9:00 to 12:00 a.m. and 1:00 to 4:30 p.m. were enough to meet the requirement for geese breeding [24]. In our study, however, the daily mating frequencies of the three goose breeds exceeded the previously reported basic mating frequency.

Reproductive performance, an important economic trait of poultry, might be associated with the intensity of sexual interaction and successful copulation has a positive impact on egg fertilization [5,6,25]. According to a previous study, the frequency of successful copulation once a day was adequate for goose reproduction [24]. In our study, we recorded that the frequencies of successful copulation of Sichuan white geese and Zhedong white geese were 2.31 and 1.94, respectively, which exceeded the basic frequency. A previous study also considered that the change in female fecundity is consistent with the daily laying rate in domestic hens [26]. Furthermore, we found that successful copulation could significantly increase both the laying and fertilization rates in Hungarian geese. Although we also found a significant correlation between successful copulation and laying rate in Hungarian geese, which was not the case in Sichuan white gees and Zhedong white geese, which reflected substantial differences in reproductive performance among the three breeds.

Typically, mate familiarity plays an important role in the reproductive process of poultry [27]. The longer time spent together with opposite sexes, the better reproductive performance of animals by increasing familiarity and enhancing cooperation [27]. Unlike commercial chickens, natural mating is the most common way accepted for female geese to breed the next generation [28]. Upon reaching sexual maturity, female geese are mixed with male geese and are allowed to reproduce in the newly formed breeding flock. However, the sexual maturity of male geese is about one month later than female geese, which results in female geese joining the breeding flock before male geese. Moreover, the findings of Black et al. that mate familiarity could improve coordination, cooperation, and responsiveness, therefore, increasing the frequency of successful copulation [29]. In our observations, we found that the time spent together with opposite sexes influenced successful copulation frequency in Zhedong white geese. Zhedong white geese which have met earlier might have more time to assess each other and thus may make a more informed choice on whether to breed together, and will also have had more time to develop compatibility that increases the frequency of successful copulation. A similar phenomenon also occurred in a free-living animal population, where animals staying together earlier potentially reduced the fitness of individuals and increased breeding success [21,30,31,32].

## 5. Conclusions

In conclusion, the percentages of Sichuan white geese and Zhedong white geese mating from 4:00 p.m. to 6:00 p.m. were over 30%, and the frequency of successful copulation was also higher than Hungarian geese. However, the successful copulation of Hungarian geese might lead to greater laying and fertilization rates. As discussed above, distinct mating characteristics were observed among the different breeds, and might have practical implications for optimal reproductive management.

## Figures and Tables

**Figure 1 animals-12-02326-f001:**
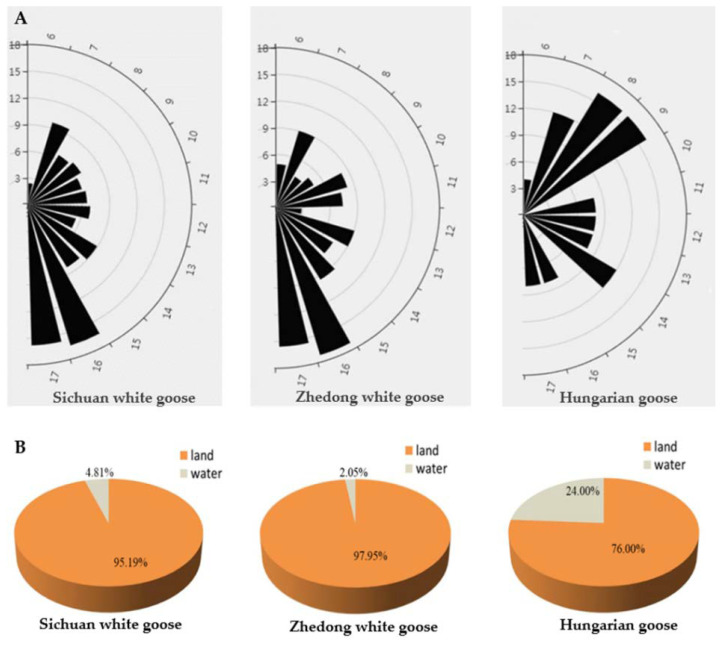
The characteristics of the mating time and location among Sichuan white geese, Zhedong white geese, and Hungarian geese. (**A**) Mating time preferences in the different goose breeds. The vertical numbers were percentages, and the numbers on the semicircle were the hours of the day. (**B**) Note: Mating location preferences in the different goose breeds. The percentages were calculated using the total number of mating over the 20-day period as the denominator.

**Figure 2 animals-12-02326-f002:**
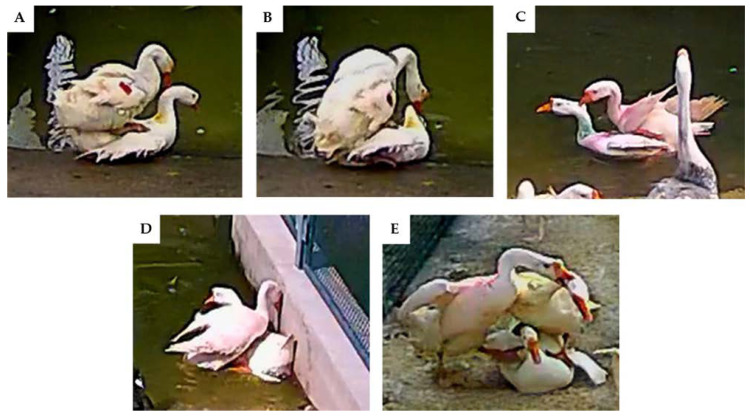
The typical mating behavior of geese. The mating behaviors were observed and recorded through the video. The typical behavior was as follows: (**A**) Mounting; (**B**) successful copulation on land; (**C**) successful copulation on water; (**D**) cooperative mating of female geese; (**E**) mating interference.

**Figure 3 animals-12-02326-f003:**
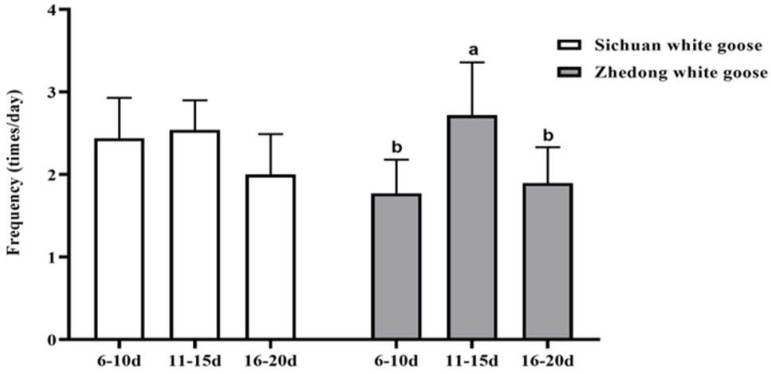
Effects of the time spent together with opposite sexes on successful copulation. The frequencies of successful copulation were assessed on days 6–10, 11–15, and 16–20 after the geese were mixed. ^a,b^ Different letters denote significant differences (*p* < 0.05).

**Table 1 animals-12-02326-t001:** Composition and nutrient content of experimental diet (air-dry basis).

Item	Content	Item	Content
Metabolic energy (MJ/kg)	11.29	Zinc (mg/kg)	33
Crude protein (%)	16	Calcium (g/kg)	45
Corn powder (g/kg)	403	Manganese (mg/kg)	55
Crushed wheat (g/kg)	250	Available phosphorus (g/kg)	35
Bean cake powder (g/kg)	135	Vitamin A (IU/kg)	4409
Green hay powder (g/kg)	127	Vitamin D3 (IU/kg)	661
Salt (g/kg)	5		

**Table 2 animals-12-02326-t002:** Description of criteria for the mating behavior.

Behavior	Definition
Mounting	The male goose approaches a crouched female goose from the front of her body and places feet on the dorsal surface of her torso.
Female cooperation	The female goose squats and lift its tail to expose the cloaca.
Successful copulation	The male goose makes cloacae contact with the female goose, and the tail was twisted underneath that of the female goose and thrust towards it.
Mating interference	The male goose interrupts the mating process by repelling, grabbing, or fighting, resulting in unsuccessful mating.

**Table 3 animals-12-02326-t003:** Frequencies of mating behaviors in the different goose breeds.

Breeds	Mounting (Times/Day)	Female Cooperation (Times/Day)	Successful Copulation (Times/Day)
Sichuan white goose	2.38 ± 0.50 ^a^	2.27 ± 0.48 ^a^	2.31 ± 0.50 ^a^
Zhedong white goose	2.01 ± 0.59 ^a^	1.83 ± 0.56 ^a^	1.94 ± 0.58 ^a^
Hungarian goose	1.15 ± 0.52 ^b^	0.82 ± 0.45 ^b^	0.89 ± 0.39 ^b^

Note: The mating behaviors were observed through the video, and the daily record from 6 a.m. to 6 p.m. during the 20-day observation period. Different letters mean a significant difference (*p* < 0.05), and the same letters mean the difference is insignificant (*p* > 0.05).

**Table 4 animals-12-02326-t004:** Correlation between successful copulation and reproduction traits.

Breeds	Laying Rate (%)	Fertilization Rate (%)
Sichuan white goose	41.67 ± 6.30 ^a^ (−0.285)	96.92 ± 2.78 ^a^ (0.691)
Zhedong white goose	27.38 ± 2.59 ^b^ (0.467)	84.63 ± 6.37 ^a^ (−0.634)
Hungarian goose	16.96 ± 3.79 ^b^ (0.985 **)	66.84 ± 9.68 ^b^ (0.992 **)

Note: Data presented are Pearson correlation coefficients. ^a,b^ Different letters denote significant differences (*p* < 0.05). ** Correlation is significant at the 0.05 level.

## Data Availability

All data generated or analyzed during this study are included in this published paper.

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
