# Peer review of "Characteristics of the Mating Behavior of Domesticated Geese from Anser cygnoides and Anser anser"

_animals, 2022, doi:10.3390/ani12182326_

Round 1
Reviewer 1 Report
Dear Author(s): The work is interesting and very beautiful. For this reason, the method section should be explained in more detail and clearly. The discussion section will be much better understood when this section is explained in detail. Correction requests are presented in the supplementary file.

Author Response
Point 1: The work is interesting and very beautiful. For this reason, the method section should be explained in more detail and clearly. The discussion section will be much better understood when this section is explained in detail. Correction requests are presented in the supplementary file.
Response 1: Thank you for your constructive comments on my manuscript. We have carefully considered the suggestion of reviewer and made the Materials and Methods section much more detailed. All of your questions were answered one by one. These changes will not influence the content and framework of the paper. Please find my itemized responses in below and my revisions in the re-submitted files.
Thanks again!
Point 2: It must be corrected.
Response 2: We apologize for the mistakes in the manuscript and also carefully checked the entire manuscript for typographic, grammatical and formatting errors. According to the reviewer’s comment, the manuscript has been revised (Line 54, page 2).
Point 3: Why was it recorded for 12 hours even though the lighting was 15 hours? It must be disclosed.
Response 3: We are grateful for the suggestion. According to the reviewer’s comment, we provided more details to describe this part as the following: “A time and location-dependent mating behavior was observed in different goose breeds. Previous study has shown that the mating time of domestic goose were primarily concentrated from 6:00 a.m. to 6:00 p.m. (Gillette, D. D. Mating and other behavior of domestic geese[J]. Appl. anim. ethol. 1977, 3, 305-319.).” We have added the information in Discussion (Lines 220-222, page 7).
Point 4: Pair-bond should be explained in more detail.
Response 4: We deeply appreciate the reviewer’s suggestion. According to the reviewer’s comment, we have added a more detailed interpretation regarding pair-bond as the following: “The first met between male and female means the beginning of pair-bond. For further understand the effects of the pair-bond duration on successful copulation, the mating behavior of 5 days were assessed on day 5, day 10, and day 15 from the beginning of the experiment (Lines 126-129, page 3).”
Point 5: What period of ovulation? is it the beginning? mid ovulation period? Information must be given. When did geese start laying eggs?
Response 5: We are extremely grateful to reviewer for pointing out this problem. And we have made some modifications in Materials and Methods. The three goose breeds, including Sichuan white geese (Anser cygnoides), Zhedong white geese (Anser cygnoides), and Hungarian geese (Anser anser), were raised separately un-der the same feeding and management regime before the start of the experiment. According to egg production records, the three goose breeds start their first egg at an average age of 210-day-old, and the study was carried out when the geese were already reproductively mature (300-day-old) (Lines 87-92, page 2). Thus, the mating behavior were observed in the timing of peak egg production.
Point 6: Have all the recordings been watched? Or was it chosen at certain times? How was the data calculated in %? These issues are important for the method.
Response 6: Thank you for the suggestion. All the recordings have been read, reviewed, and discussed by researchers. As described in the Behavior Observation of our article. The mating behaviors of all geese were recorded for 12 hours per day from 6:00 a.m. to 6:00 p.m. over the 20 days observation period (Lines 114-115, page 3). Therefore, a total of 20 days of data in behavior trials period were included in the statistical analysis to ensure data quality and completeness. Furthermore, the percentages were calculated using the total number of mating as the denominator. We have made revisions in the manuscript accordingly (Lines 176-177, page 5).
Point 7: Information about the process of turning and cooling the eggs should also be given. Also, has water spraying been done? If it was done, how many degrees of water was it made?
Response 7: Our deepest gratitude goes to you for your careful work and thoughtful suggestions that have helped improve this paper substantially. This section was revised and modified according to the information showed in the manuscript suggested by the reviewer. Before hatching, the eggs were cleaned and sterilized with 70% ethanol (Lines 135-136, page 4). And then the eggs were incubated under standard conditions. All the eggs were turned every two hours until 28 days (Lines 136-137, page 4). In this experiment, rather than using the spraying water, we opened the incubator to reduce temperature of eggs. The incubator should be opened twice a day for 35 minutes each time after the 17th day of incubation until all eggs have hatched, and the eggshells’ temperature were regulated at 30-35℃(Lines 140-142, page 4).
Finally, we really appreciate all your comments and suggestions. Thanks very much for taking your time to review this manuscript. Those comments are all valuable and very helpful for revising and improving our paper. We have revised the manuscript accordingly, and our point-by-point responses are presented above.

Reviewer 2 Report
The manuscript fits well within the scope of the journal. The Authors have investigated an interesting topic and the theme has been properly described. Objectives of the study were clearly defined.
The Introduction is written concisevely but provides sufficient background. The methods have been properly described, the design of the experiment and statistical methods applied allow to make reliable conclusions.
Results are well presented and thoroughly discussed and data interpretation is appropriate.
The manuscript is well written, presented and discussed, and understandable to a specialist readership.
No significant limitations have been detected, whereas the paper presents novel and useful findings. The results have significant practical implications.
Specific comments:
- the authors should explain why Hungarian goose was selected for comparisons with Chinese breeds. There are many European goose breeds and it is not clear why Hungarian goose was selected. Is it a breed commonly kept in China? Were individuals coming from breeding stock adapted to Chinese climatic conditions used in the experiments or were they imported just prior to the experiment? In the Discussion, the authors should avoid generalization if they only tested one European goose breed they should not assume all European breeds would behave same.
Author Response
Point 1: The manuscript fits well within the scope of the journal. The Authors have investigated an interesting topic and the theme has been properly described. Objectives of the study were clearly defined. The Introduction is written concisely but provides sufficient background. The methods have been properly described, the design of the experiment and statistical methods applied allow to make reliable conclusions. Results are well presented and thoroughly discussed and data interpretation is appropriate. The manuscript is well written, presented and discussed, and understandable to a specialist readership. No significant limitations have been detected, whereas the paper presents novel and useful findings. The results have significant practical implications.
Response 1: Thanks very much for taking your time to review this manuscript. We really appreciate the reviewer’s positive evaluation of our work. According to your advice, we amended the relevant part in manuscript. All of your questions were answered one by one, and hope that the correction will meet with approval.
Once again, thank you very much for your comments and suggestions.
Point 2: The authors should explain why Hungarian goose was selected for comparisons with Chinese breeds. There are many European goose breeds and it is not clear why Hungarian goose was selected. Is it a breed commonly kept in China? Were individuals coming from breeding stock adapted to Chinese climatic conditions used in the experiments or were they imported just prior to the experiment?
Response 2: We are extremely grateful to reviewer for pointing out this problem. According to the reviewer’s comment, we have added a more detailed explain why Hungarian goose was selected for comparisons with Chinese breeds. China is the world’s largest goose farming country and has a wide variety of goose resources. In recent years, Hungarian geese were widely raised as an excellent introduced breed and their popularity was increasing in many regions of China. However, geese are accompanied by a variety of different reproductive performance among breeds. For these reasons, Hungarian goose was selected for comparisons with Chinese breeds (Sichuan white goose and Zhedong white goose). Moreover, Hungarian geese had been raised under the same environment as Chinese geese since they were introduced into China, and adapted to the climatic conditions. We have added the suggested content to the revised manuscript (Lines 52-55, page 2).
Point 3: In the Discussion, the authors should avoid generalization if they only tested one European goose breed they should not assume all European breeds would behave same.
Response 3: We agree with the reviewer’s point. We have made correction according to the reviewer’s comments (Lines 225, 226 and 252, page 7).
Finally, we really appreciate all your comments and suggestions. Thanks very much for taking your time to review this manuscript. Those comments are all valuable and very helpful for revising and improving our paper. We have revised the manuscript accordingly, and our point-by-point responses are presented above.
